# Elastic Strain Relaxation of Phase Boundary of α′ Nanoscale Phase Mediated via the Point Defects Loop under Normal Strain

**DOI:** 10.3390/nano13030456

**Published:** 2023-01-22

**Authors:** Zhengwei Yan, Shujing Shi, Peng Sang, Kaiyue Li, Qingqing Qin, Yongsheng Li

**Affiliations:** 1School of Materials Science and Engineering, Nanjing University of Science and Technology, Nanjing 210094, China; 2MIIT Key Laboratory of Advanced Metallic and Intermetallic Materials Technology, Nanjing 210094, China

**Keywords:** point defects, nanophase, normal strain, morphology, simulation

## Abstract

Irradiation-induced point defects and applied stress affect the concentration distribution and morphology evolution of the nanophase in Fe–Cr based alloys; the aggregation of point defects and the nanoscale precipitates can intensify the hardness and embrittlement of the alloy. The influence of normal strain on the coevolution of point defects and the Cr-enriched α′ nanophase are studied in Fe-35 at.% Cr alloy by utilizing the multi-phase-field simulation. The clustering of point defects and the splitting of nanoscale particles are clearly presented under normal strain. The defects loop formed at the α/α′ phase interface relaxes the coherent strain between the α/α′ phases, reducing the elongation of the Cr-enriched α′ phase under the normal strains. Furthermore, the point defects enhance the concentration clustering of the α′ phase, and this is more obvious under the compressive strain at high temperature. The larger normal strain can induce the splitting of an α′ nanoparticle with the nonequilibrium concentration in the early precipitation stage. The clustering and migration of point defects provide the diffusion channels of Cr atoms to accelerate the phase separation. The interaction of point defect with the solution atom clusters under normal strain provides an atomic scale view on the microstructure evolution under external stress.

## 1. Introduction

An irradiation cascade reaction produces point defects in crystal materials. Point defects are either intrinsic, as with vacancies, interstitial atoms, and their small clusters, or extrinsic, as with impurities and dopants; they play a major role in materials’ properties and microstructure evolution. Some properties of these point defects, such as their formation and migration energies, are mainly determined by the region in the immediate vicinity of defects where the crystal structure is strongly disturbed [1]. The formation of point defects not only promotes the phase separation [2,3], but also produces the lattice misfit and the elastic energy [4] to affect the morphology of the precipitated phase.

Furthermore, point defects can induce a long-range perturbation of the host lattice, leading to an elastic interaction with other structural defects, impurities, or an applied elastic field. Following the work of Eshelby et al. [5], the elastic model of a point defect can be regarded as a spherical inclusion that is forced into a spherical hole of slightly different size in an infinite elastic medium. The variations in the “size” and “shape” of point defects with the strain field is described by assigning different elastic constants to the inclusion. In the study of Clouet et al. [1], the elastic model was used to describe the evolution of a point defect in an external elastic field, and the influence of a defect was discussed using the dipole tensors.

The strain produced by the thermal or external stress has a substantial effect on the phase separation and spatial morphology [6,7]. In the study of a monolayer alloy, the strain can induce a structural phase transformation [8]. Furthermore, applied tensile or compressive stresses can induce the orientation morphologies of precipitates [9,10], and directional coarsening is observed when annealed with tensile and compressive stresses [11,12]. Prikhodko and Ardell found that applied compressive stress retards the growth of the precipitates and widens the distribution of particle size [13,14]. However, the interaction between the point defects and the nanophase under the applied strain is still not clear; also, the morphology of the defects cluster and its effect on the evolution of nanophase are not studied.

Fe–Cr-based stainless steels are used as the critical material in nuclear power plants for their excellent resistance to oxidation, corrosion resistance, and mechanical properties; both the Fe-enriched α and the Cr-enriched α′ phase with bcc structure are formed via phase separation under high temperature, irradiation, or stress. Tong et al. studied the morphology and kinetics evolution of the Cr-enriched α′ nanophase under multiaxial strain by utilizing the three-dimensional phase-field simulation; they found that phase separation is accelerated under unidirectional normal strain, and is only slowed down under triaxial compressive strain [15]. When the precipitate’s size is at the nanoscale, the irradiation point defects, such as vacancy and interstitial atoms, will essentially affect the atoms’ migrations, and then the formation of a Cr-enriched α′ phase will be faster [16,17]. These Cr-enriched α′ nanoscale precipitates can prevent the motion of dislocations, which causes an increase in hardness [18]; thus, unfavorable thermal embrittlement will be produced [19]. However, the evolutions of nanoparticles and point defects in Fe–Cr alloys under normal stress and radiation are not clarified. Therefore, the effect of strain accompanied with irradiated defects on the precipitation is an interesting question.

In addition, temperature influences the phase separation under strain and irradiation. Zhu et al. found that the effect of applied strain is more obvious at high temperature [20], indicating that the initial stage phase separation is faster and that the elongation of the particle is obvious at high temperature under applied strain. The strain can induce the splitting of coherent precipitates in Ni-based superalloys close to the solvus temperature [21,22], and the particle splitting based on the competition of the coherency elastic energy and the interfacial energy was discussed by Lee [23]. This splitting of the γ′ particle is caused by the stress relaxation of the internal misfit strain between the γ’ and γ phases, and an applied load affects the splitting direction [24]. However, the splitting of the particle with the influence of point defects under the difference temperature is not well studied. Therefore, the effect of temperature on the evolution of defects and the splitting of nanoparticles needs further study in the strained system.

In view of previous studies regarding the irradiation defects and strains in phase separation, the evolution of point defects and nanophase under the normal strain will be studied in this work. The phase-field model is developed coupling with the production and recombination of vacancy and interstitial atoms, and the external strains with the coherent strain are incorporated. The effects of normal strain on the concentration evolution of point defects and Cr in the Cr-enriched nanophase, and the shape change of the nanophase around the defects loop are revealed. In addition, the splitting of nanoparticles with point defects is presented with the temperature change.

## 2. Model and Methods

### 2.1. Elastic Strain Including Vacancy and Interstitial Atoms

The strain $\varepsilon_{ij}^{a}$ caused by the external stress influences the morphology of the nanophase, which is introduced as part of the elastic strain. Therefore, the elastic strain can be calculated by [25]
(1)$$\varepsilon_{ij}^{\mathrm{el}}=\varepsilon_{ij}^{a}+\varepsilon_{ij}-\varepsilon_{ij}^{0}$$
where $\varepsilon_{ij}$ is the heterogeneous strain, and $\varepsilon_{ij}^{0}$ is the eigenstrain caused by the compositional heterogeneity and point defects and can be calculated by [4]
(2)$$\varepsilon_{ij}^{0}=\delta_{ij}\left(\varepsilon_{0}\Delta c+\varepsilon_{V0}\Delta c_{V}+\varepsilon_{I0}\Delta c_{I}\right)$$
where $\delta_{ij}$ is the Kronecker-delta function. $\varepsilon_{0}=\frac{1}{a}\left(\frac{da}{dc_{\mathrm{Cr}}}\right)$, $\varepsilon_{V0}=\frac{1}{a}\left(\frac{da}{dc_{V}}\right)$ and $\varepsilon_{I0}=\frac{1}{a}\left(\frac{da}{dc_{I}}\right)$ are the expansion coefficients of the lattice parameter due to the introduction of concentration inhomogeneity, vacancy, and interstitial atoms [4], $\Delta c=c_{\mathrm{Cr}}-c_{0}$, where $c_{\mathrm{Cr}}$ is the concentration of Cr, $c_{0}$ is the initial average concentration of Cr, $\Delta c_{m}=c_{m}-c_{m0}\;\left(m=V,I\right)$, $c_{m}$ is the concentration of vacancy and interstitial atoms, and $c_{m0}$ is the initial average concentration of vacancy and interstitial atoms and can be calculated via [26]
(3)$$c_{m0}=\exp(-\frac{E_{m}^{f}}{RT})$$
where $E_{m}^{f}$ is the formation energy of defect. The vacancy and interstitial atoms are assumed to occur concurrently; therefore, the concentrations of solution atoms and point defects satisfy $c_{\mathrm{Cr}}+c_{V}+c_{I}+c_{\mathrm{Fe}}=1.0$; $c_{\mathrm{Fe}}$ is the concentration of Fe. The variation of the stress-free lattice parameter is $a=\left(1-c_{\mathrm{Cr}}-c_{V}-c_{I}\right)a_{\mathrm{Fe}}+c_{\mathrm{Cr}}a_{\mathrm{Cr}}$, where $a_{\mathrm{Cr}}$ and $a_{\mathrm{Fe}}$ are the lattice parameters of Cr and Fe atoms. The elastic strain can be calculated using Hook’s law [27]
(4)$$\sigma_{ij}^{\mathrm{el}}=C_{ijkl}\varepsilon_{ij}^{\mathrm{el}}=\left(C_{ijkl}^{0}+\Delta C_{ijkl}\Delta c\right)\left(\varepsilon_{ij}^{a}+\varepsilon_{ij}-\varepsilon_{ij}^{0}\right)$$
where $C_{ijkl}^{0}$ is the average elastic modulus tensors and can be calculated by
(5)$$C_{ijkl}^{0}=\lambda C_{ijkl}^{P}+\left(1-\lambda \right)C_{ijkl}^{M}$$
where λ is the volume fraction of precipitates, and $C_{ijkl}^{P}$ and $C_{ijkl}^{M}$ are the elastic modulus tensors of precipitates and matrix, respectively. $\Delta C_{ijkl}$ is the difference of the elastic modulus tensors and is given by
(6)$$\Delta C_{ijkl}=C_{ijkl}^{P}-C_{ijkl}^{M}$$

In the Fe–Cr alloy system, the elastic constants of the precipitates are $C_{11}^{P}=3.5\times 10^{11}\mathrm{Pa}$, $C_{12}^{P}=0.678\times 10^{11}\mathrm{Pa}$ and $C_{44}^{P}=1.008\times 10^{11}\mathrm{Pa}$, and those of the matrix are $C_{11}^{M}=2.33\times 10^{11}\mathrm{Pa}$, $C_{12}^{M}=1.354\times 10^{11}\mathrm{Pa}$ and $C_{44}^{M}=1.178\times 10^{11}\mathrm{Pa}$ [28]. The elastic strain is related to the displacement
(7)$$\varepsilon_{kl}=\frac{1}{2}\left(\frac{\partial u_{k}}{\partial x_{l}}+\frac{\partial u_{l}}{\partial x_{k}}\right)$$

Since the mechanical equilibrium with respect to elastic displacements is established much faster than through any diffusion processes, for any given distribution of composition, the system is always at mechanical equilibrium [29].
(8)$$\frac{\partial \sigma_{ij}^{\mathrm{el}}}{\partial x_{j}}=0$$

Then, the elastic strain energy density can be calculated by
(9)$$E_{e}=\frac{1}{2}C_{ijkl}\left(\varepsilon_{ij}^{a}+\varepsilon_{ij}-\varepsilon_{ij}^{0}\right)\left(\varepsilon_{kl}^{a}+\varepsilon_{kl}-\varepsilon_{kl}^{0}\right)$$

### 2.2. Free Energy

The total free energy [30] includes the chemical free energy, gradient energy, and elastic energy
(10)$$F\left(c_{\mathrm{Cr}},c_{V},c_{I}\right)=\int_{V}\left\{\begin{array}{l} \frac{1}{V_{m}}\left[G\left(c_{\mathrm{Cr}},c_{V},c_{I},T\right)+\frac{k_{\mathrm{Cr}}}{2}{\left|\nabla c_{\mathrm{Cr}}\right|}^{2}+G_{\mathrm{mag}}\right]\\ +\frac{k_{V}}{2}{\left|\nabla c_{V}\right|}^{2}+\frac{k_{I}}{2}{\left|\nabla c_{I}\right|}^{2}+E_{e}\left(c_{\mathrm{Cr}},c_{V},c_{I},T\right) \end{array}\right\}dV$$
where $G\left(c_{\mathrm{Cr}},c_{V},c_{I},T\right)$ is the molar Gibbs energy, chemical free density is $E_{c}=G/V_{m}$, and $V_{m}$ is the molar volume. $k_{\mathrm{Cr}}$ is the gradient energy coefficients for the α/α′ phase, where $k_{\mathrm{Cr}}=\frac{1}{6}{r_{0}}^{2}L_{\mathrm{FeCr}}$, and $L_{\mathrm{FeCr}}$ is the interaction parameter between Fe and Cr, where $L_{\mathrm{FeCr}}=20500-9.68T$J mol^−1^ [31]. $r_{0}$ is the interatomic distance and changes with concentration. $k_{V}$ and $k_{I}$ are the gradient energy coefficients for vacancy and interstitial atoms, where $k_{V}=k_{I}=6.91\times 10^{-9}\;{J\;m}^{-1}$ [4]. Then, the molar Gibbs energy can be written as
(11)$$\begin{array}{c} G\left(c_{\mathrm{Cr}},c_{V},c_{I},T\right)=E_{V}^{f}c_{V}+E_{I}^{f}c_{I}+c_{\mathrm{Fe}}G_{\mathrm{Fe}}^{0}+c_{\mathrm{Cr}}G_{\mathrm{Cr}}^{0}+L_{\mathrm{FeCr}}c_{\mathrm{Cr}}c_{\mathrm{Fe}}\\ +RT\left[\begin{array}{l} c_{\mathrm{Cr}}\ln c_{\mathrm{Cr}}+c_{V}\ln c_{V}+c_{I}\ln c_{I}+\\ \left(1-c_{\mathrm{Cr}}-c_{V}-c_{I}\right)\ln\left(1-c_{\mathrm{Cr}}-c_{V}-c_{I}\right) \end{array}\right] \end{array}$$
where $G_{\mathrm{Fe}}^{0}$ and $G_{\mathrm{Cr}}^{0}$ are the molar Gibbs free energies of the element of Fe and Cr, respectively. $E_{V}^{f}$ and $E_{I}^{f}$ are the respective formation energies of vacancy and interstitial atoms, which are taken as 2.24 eV and 3.03 eV [32]. *G*_mag_ represents the magnetic ordering contribution to the Gibbs free energy [33]
(12)$$G_{\mathrm{mag}}=RT\ln\left(\beta +1\right)f\left(\tau \right)$$
where β is the atomic magnetic moment in the Bohr magneton, dependent on the concentration of Cr, $\beta =2.22c_{\mathrm{Fe}}-0.008c_{\mathrm{Cr}}-0.85c_{\mathrm{Fe}}c_{\mathrm{Cr}}$, and f(τ) satisfies the polynomial function as follows [33]
(13)$$f\left(\tau \right)=\left\{ \begin{array}{l} -0.90530\tau^{-1}+1.0-0.153\tau^{3}-6.8\times 10^{-3}\tau^{9}-1.53\times 10^{-3}\tau^{15},\tau <1\\ -0.06417\tau^{-5}-2.037\times 10^{-3}\tau^{-15}-40278\times 10^{-4}\tau^{-25},\tau \geq 1 \end{array}\right.$$

### 2.3. Phase-Field Model

The effect of irradiation on phase separation is accomplished via the defects action. So, the field variations in the phase-field model are described by the concentration field of Cr $c_{\mathrm{Cr}}\left(r,t\right)$, vacancy $c_{V}\left(r,t\right)$, and interstitial atoms $c_{I}\left(r,t\right)$ through the Cahn–Hilliard diffusion equation [34,35]
(14)$$\frac{\partial c_{\mathrm{Cr}}\left(r,t\right)}{\partial t}=V_{m}\nabla \left(M\nabla \frac{\delta F}{\delta c_{\mathrm{Cr}}}\right)+\xi_{\mathrm{Cr}}$$
(15)$$\frac{\partial c_{V}\left(r,t\right)}{\partial t}=V_{m}\nabla \left(M_{V}\nabla \frac{\delta F}{\delta c_{V}}\right)+\xi_{V}+P_{V}-R_{\mathrm{VI}}-S_{V}$$
(16)$$\frac{\partial c_{I}\left(r,t\right)}{\partial t}=V_{m}\nabla \left(M_{I}\nabla \frac{\delta F}{\delta c_{I}}\right)+\xi_{I}+P_{I}-R_{\mathrm{VI}}-S_{I}$$
where r and *t* are the spatial coordinate and time, respectively, and M is the chemical mobility of the Fe–Cr alloy and is given by Darken’s equation [36]
(17)$$M=\left(c_{\mathrm{Cr}}M_{\mathrm{Fe}}+c_{\mathrm{Fe}}M_{\mathrm{Cr}}\right)c_{\mathrm{Cr}}c_{\mathrm{Fe}}$$
where $M_{i}\;\left(i=\mathrm{Fe},\mathrm{Cr}\right)$ is the atomic mobility of Fe or Cr, and can be expressed as $M_{i}=\frac{D_{i}}{RT}$, where $D_{i}\;\left(i=\mathrm{Fe},\mathrm{Cr}\right)$ is the diffusion coefficient of Fe or Cr, *R* is the gas constant, and *T* is the absolute temperature. $M_{V}$ and $M_{I}$ are the chemical mobilities of vacancy and interstitial atoms, respectively, which can be described by $M_{j}=\frac{c_{j}D_{j}}{RT}\;\left(j=V,I\right)$, where $D_{j}\;\left(j=V,I\right)$ is the diffusion coefficient of vacancy or interstitial atoms. $\xi_{i}\;\left(i=\mathrm{Cr},V,I\right)$ are the thermal fluctuations to introduce the random distribution of concentration and point defects to trigger the nucleation, and their magnitudes are set as $10^{-8}$. $P_{V}$ and $P_{I}$ are the production rates of vacancy and interstitial atoms, respectively, which are taken as $1\times 10^{-8}\;\mathrm{dpa}/s$ [37]. $R_{\mathrm{VI}}$ is the recombination rate of the vacancy and interstitial atoms [4,38], and is given by
(18)$$R_{\mathrm{VI}}=\frac{4\pi r_{\mathrm{VI}}\left(D_{V}+D_{I}\right)}{\Omega }c_{V}c_{I}$$
where $r_{\mathrm{VI}}$ is the recombination radius and is equivalent to the atomic radius, $r_{\mathrm{VI}}=\frac{\sqrt{3}a}{4}$. Ω is the atomic volume. $S_{V}$ and $S_{I}$ are the sink rates of the vacancy and interstitial atoms, respectively.

### 2.4. Numerical Calculation

By substituting Equation (10) into Equations (14)–(16) respectively, the evolution equations of the solution atoms and point defects are given by
(19)$$\frac{\partial c_{\mathrm{Cr}}\left(r,t\right)}{\partial t}=\nabla \left[M\nabla \left(\frac{\delta G}{\delta c_{\mathrm{Cr}}}-k_{\mathrm{Cr}}\nabla^{2}c_{\mathrm{Cr}}+V_{m}\frac{\delta E_{e}}{\delta c_{\mathrm{Cr}}}\right)\right]+\xi_{\mathrm{Cr}}$$
(20)$$\frac{\partial c_{V}\left(r,t\right)}{\partial t}=\nabla \left[M_{V}\nabla \left(\frac{\delta G}{\delta c_{V}}-V_{m}k_{V}\nabla^{2}c_{V}+V_{m}\frac{\delta E_{e}}{\delta c_{V}}\right)\right]+\xi_{V}+P_{V}-R_{\mathrm{VI}}-S_{V}$$
(21)$$\frac{\partial c_{I}\left(r,t\right)}{\partial t}=\nabla \left[M_{I}\nabla \left(\frac{\delta G}{\delta c_{I}}-V_{m}k_{I}\nabla^{2}c_{I}+V_{m}\frac{\delta E_{e}}{\delta c_{I}}\right)\right]+\xi_{I}+P_{I}-R_{\mathrm{VI}}-S_{I}$$

For solving Equations (19)–(21) numerically, the following normalization parameters are introduced, $r^{*}=r/l$, $t^{*}=tD/l^{2}$, $M_{i}^{*}=RT_{0}M_{i}/D$, $G^{*}={G/RT}_{0}$, $k_{\mathrm{Cr}}^{*}=k_{\mathrm{Cr}}/RT_{0}l^{2}$, $k_{V}^{*}=V_{m}k_{V}/RT_{0}l^{2}$, $k_{I}^{*}=V_{m}k_{I}/RT_{0}l^{2}$, $E_{c}^{*}={V_{m}E_{c}/RT}_{0}$, $E_{e}^{*}={V_{m}E_{e}/RT}_{0}$, $\nabla^{*}=\frac{\partial }{\partial r^{*}}=\frac{\partial }{\partial \left(r/l\right)}$, $\xi_{i}^{*}=\frac{l^{2}}{D}\xi_{i}$, $P_{i}^{*}=\frac{l^{2}}{D}P_{i}$, $R_{i}^{*}=\frac{l^{2}}{D}R_{i}$, $S_{i}^{*}=\frac{l^{2}}{D}S_{i}$, where *l* is the grid length of the simulation cell and is set as the average lattice parameter of Cr and Fe, $l=0.29\times 10^{-9}\;m$. $D=10^{-26}{\;m}^{2}{\;s}^{-1}$ is the diffusion coefficient and is used for dimensionless, and $T_{0}=$900 K is the critical temperature of spinodal decomposition of the Fe–Cr alloy [39]. The dimensionless grid size is $\Delta x^{*}=\Delta y^{*}=1.0$, and the 2D simulation cell is $64\Delta x^{*}\times 64\Delta y^{*}$ with the grid length Δx=Δy=l. The time step is Δt*=0.001. The semi-implicit Fourier spectrum method is used to solve Equations (19)–(21) with a periodic boundary condition. Barker et al. [40] concluded that the qualitative morphology is very similar between the 2D and 3D simulations when they discussed the effect of concentration on the spinodal decomposition of the Fe–Cr alloys. Zhu et al. also showed that the morphology and the dynamic evolution law of the α′ phase are similar for the 2D and 3D simulations under the uniaxial strain [41], although the elasticity is inherently a 3D problem. In addition, in the microstructure evolution of Ni-base superalloys during [110] creep loading, the 3D simulation focuses on the microstructure formation, and more quantitative analyses are performed in 2D simulation [42]. Therefore, the 2D simulation can provide reasonable results and save more time in simulation.

## 3. Results and Discussion

The simulation starts from a particle with an initial concentration of 35 at.% Cr in a circle region with a radius of 5 *l* and 25 at.% Cr in the other regions of the simulation cell, which is to ensure that there is only one particle nucleation in the system, and that the vacancy and interstitial atoms are randomly distributed in the simulation cell. The input concentrations of vacancy and interstitial atoms are $c_{V}^{i}=c_{I}^{i}=0$ and $c_{V}^{i}=c_{I}^{i}=1\times 10^{-7}$ for comparing the effects with and without defects.

### 3.1. α′ Phase Evolution with a Point Defects Loop under Normal Strain

In this section, the effects of normal strain on the concentration evolution of Cr in the Cr-enriched α′ phase with the vacancy and interstitial atoms at 750K are studied. The tensile and compressive strains show different results on the elongation of the α′ nanophase, and are explained by the distribution of the elastic energy density in the system with defects loop and nanophase.

Figure 1 shows the morphology of the Cr-enriched α′ nanophase and vacancy loop at 750 K under normal trains $\varepsilon_{xx}^{a}=0$, $\varepsilon_{xx}^{a}=-0.025$, and $\varepsilon_{xx}^{a}=0.025$. Under the effect of compressive strain $\varepsilon_{xx}^{a}=-0.025$ along the *x**-axis, the α′ phase particle elongates along the *x**-axis, as shown in Figure 1b. When the tensile strain $\varepsilon_{xx}^{a}=0.025$ is along the *x**-axis, the α′ phase particle is elongated along the *y**-axis, as shown in Figure 1c. This orientation of the α′ phase depends on the sign of the eigenstrain and the applied strain [43]. The elongation direction under the tensile or compressive strain depends on whether the phase is hard or soft, and the lattice mismatch of the precipitates is either positive or negative [44]. In the Fe–Cr alloy, the α′ phase is the soft phase and the lattice mismatch is the positive swelling coefficient [20], and so the orientation of the α′ phase is perpendicular to the direction of tensile strain and parallel to the compression strain direction [15,45].

When the point defects loop forms at the α/α′ phase boundaries, as shown in Figure 1g–i, the coherent relationship between the α and α′ phases is released, and the egenstrain is reduced, so that the elongation of the Cr-enriched particle along the tensile direction is weakened under normal strain, as shown in Figure 1e,f. In the elastic model of point defect [1], the relation of defect and elastic dipole is established via Kanzaki forces. Kanzaki forces are applied to the atoms in the neighborhood of the point defect. Taking vacancy as an example, when the atom is added in the vacancy reversely, a force is produced between the host atom and the neighbor atoms, which is contrary to the Kanzaki forces. So, the point defect offsets the force between the neighbor atoms of the normal crystal lattice, and the strain between the neighbor atoms of Fe and Cr is released. As a result, the elongation of the particle around the defects loop is restricted.

To identify the particle shape change under the influences of normal strain and defects loop, the distributions of local Cr concentration, *c*_Cr_, are shown in Figure 2 along *x** = 32 and *y** = 32, where, the *L^x^*^*^ and *L^y^*^*^ delegate the concentration width for *c*_Cr_ = 0.5 in the *x**-axis and *y**-axis directions, respectively. Then, the aspect ratio *k* of the α′ phase is calculated using the ratio of the particle′s length to width, i.e., *k* = *L^x^*^*^/*L^y^*^*^ and *k* = *L^y^*^*^/*L^x^*^*^ under the compressive and tensile strains, respectively. When *t** = 50, the concentration of Cr reaches the equilibrium value 0.84 and 0.83 with and without strain, as shown in Figure 2.

As the initial concentration of point defect increases from 0.0 to 1.0 × 10^–7^, when $\varepsilon_{xx}^{a}=0$, the aspect ratio *k =* 1 of the α′ phase, as shown in Figure 2a,d. When $\varepsilon_{xx}^{a}=-0.025$, the aspect ratio decreases from 4.54 to 2.22 as the raising of the point defect concentration, as shown in Figure 2b,e. The value of *k* decreases from 1.71 to 1.37 with the effect of the tensile strain $\varepsilon_{xx}^{a}=0.025$, as shown in Figure 2c,f. Therefore, when there is a defects loop at the interface of the α/α′ phase, the deformation of the Cr-enriched nanoparticle is weakened under the compressive or tensile strain.

According to the results of rafting in the superalloys, the elastic energy is dominant in the formation of the rafting morphology [11]. Therefore, the distribution of elastic energy can clearly show the influence of strain on particle deformation with the point defects. Figure 3 shows the dimensionless elastic energy density $E_{e}^{*}$ distribution along the direction of the particle elongation through the center of the α′ phase under different strains at 750 K. Points A and B show the peak values of the dimensionless elastic energy density $E_{\mathrm{eA}}^{*}$ and $E_{\mathrm{eB}}^{*}$ at the interface regions of the α/α′ phase, respectively.

When $\varepsilon_{xx}^{a}=0$, the $E_{e}^{*}$ with defects is higher than that without defects, and the peak value of the dimensionless elastic energy density increases from $E_{\mathrm{eA}}^{*}=1.4\times 10^{-3}$ to $E_{\mathrm{eB}}^{*}=2.7\times 10^{-3}$ at the interface region, as shown in Figure 3a. The result indicates that, when there is no normal strain, the $E_{e}^{*}$ at the interface region is elevated when there is a defects loop, even if the coherent eigenstrain between the α/α′ phase is reduced. However, the vacancy or interstitial atom produces the negative eigenstrain to increase the elastic strain. The large $E_{e}^{*}$ results in a higher concentration of Cr under $\varepsilon_{xx}^{a}=0$, as shown in Figure 4a,b.

Under the compressive strain $\varepsilon_{xx}^{a}=-0.025$, when the concentration of point defects increases to 1.0 × 10^–7^, the peak value of the dimensionless elastic energy density of the interface regions also increases from $E_{\mathrm{eA}}^{*}=3.7\times 10^{-2}$ to $E_{\mathrm{eB}}^{*}=5.2\times 10^{-2}$, as shown in Figure 3b. Compared with Figure 3a and Figure 3c, the $E_{e}^{*}$ in the α′ phase is higher than that without normal strain and lower than that with tensile strain. In addition, the $E_{e}^{*}$ in the matrix is higher than that of the α′ phase under the compressive strain $\varepsilon_{xx}^{a}=-0.025$. For soft precipitates, the energy required for a sheet or a lathy shape deformation is lower than that of spherical precipitates, and the matrix will prevent them from contracting along a sheet or a lathy shape [9]. In the Fe–Cr alloy, the α′ phase is softer than the matrix, so the lamellar and lathy shapes are formed obviously under uniaxial compression strain, and the aspect ratio *k* of the α′ phase is greater than that of the tensile strain (see Figure 2).

Under tensile strain $\varepsilon_{xx}^{a}=0.025$, the peak values $E_{\mathrm{eA}}^{*}$ and $E_{\mathrm{eB}}^{*}$ at the interface of the α/α′ phase are similar with and without the defects loop, as shown in Figure 3c. The $E_{e}^{*}$ inside the particle is decreased as the concentration of point defects increases to $c_{V}^{i}=c_{I}^{i}=1\times 10^{-7}$, which is contrary to that of $\varepsilon_{xx}^{a}=0$ and $\varepsilon_{xx}^{a}=-0.025$. Under tensile strain, the low $E_{e}^{*}$ inside the α′ phase weakens the deformation of the particles.

### 3.2. Concentration Evolution of the α′ Phase with the Effects of Point Defects, Strains and Temperature

The concentration evolution of Cr in the Cr-enriched α′ phase with the vacancy and interstitial atoms are studied under normal strain at 700 K and 750 K; the influences of temperature, strain, and vacancy on equilibrium concentration are analyzed. The concentration distributions of Cr in the α′ phase and vacancy at 750 K are shown in Figure 4. The concentration changes of Cr at 750 K under the effects of compressive and tensile strains are listed in Table 1. When there are no point defects, the Cr concentration in the α′ phase increases from $c_{\mathrm{Cr}}^{0}=0.48$ to $c_{\mathrm{Cr}}^{c}=0.49$ under the effect of compressive strain $\varepsilon_{xx}^{a}=-0.025$, and to $c_{\mathrm{Cr}}^{t}=0.58$ under tensile strain $\varepsilon_{xx}^{a}=0.025$, as shown in Figure 4a. The concentration changes of Cr under the effect of compressive and tensile strain are $\Delta c_{\mathrm{Cr}}=c_{\mathrm{Cr}}^{c}-c_{\mathrm{Cr}}^{0}=0.01$ and $\Delta c_{\mathrm{Cr}}=c_{\mathrm{Cr}}^{t}-c_{\mathrm{Cr}}^{0}=0.10$, respectively. The relative concentration differences, $\Delta c_{\mathrm{Cr}}/c_{\mathrm{Cr}}^{0}$, are 0.021 and 0.208, respectively. The results indicate that the effect of compressive strain on initial concentration clustering is smaller than that of tensile strain when there are no point defects.

When the initial point defect concentration is $c_{V}^{i}=c_{I}^{i}=1\times 10^{-7}$, the concentration of Cr in the α′ phase increases from $c_{\mathrm{Cr}}^{0}=0.55$ to $c_{\mathrm{Cr}}^{c}=0.64$ and $c_{\mathrm{Cr}}^{t}=0.73$ under the effects of compressive strain $\varepsilon_{xx}^{a}=-0.025$ and tensile strain $\varepsilon_{xx}^{a}=0.025$, as shown in Figure 4b. The concentration changes of Cr in the α′ phase are $\Delta c_{\mathrm{Cr}}=0.09$ and $\Delta c_{\mathrm{Cr}}=0.18$, and the relative concentration differences $\Delta c_{\mathrm{Cr}}/c_{\mathrm{Cr}}^{0}$ are 0.164 and 0.327, respectively. Compared with no defects, the $\Delta c_{\mathrm{Cr}}$ increases by 0.08 under both $\varepsilon_{xx}^{a}=-0.025$ and $\varepsilon_{xx}^{a}=0.025$, and the relative concentration difference $\Delta c_{\mathrm{Cr}}/c_{\mathrm{Cr}}^{0}$ increases by 0.143 and 0.119 under the effects of compression and tensile strains. The results show that the point defects promote the Cr atoms cluster, and the effect of compressive strain on accelerating the phase separation is enhanced with the point defects. However, the effect of strain is slight on the concentration change of point defects, as shown in Figure 4d.

A previous study of Fe–Cr alloy shows that the effect of point defects on the phase separation is reduced at low temperature [17]. The concentration distributions of Cr and vacancy at 700 K under different strains are displayed in Figure 5. The concentration changes of Cr with and without defect at 700 K under the effects of compressive and tensile strains are listed in Table 2, where, the peak values of *c*_Cr_ are defined as $c_{\mathrm{Cr}}^{i0}$ (*i* = c, t) for $c_{V}^{i}=c_{I}^{i}=0$ and $c_{\mathrm{Cr}}^{i1}$ for $c_{V}^{i}=c_{I}^{i}=1\times 10^{-7}$ under the effect of compressive (*i* = c) $\varepsilon_{xx}^{a}=-0.025$ and tensile strains (*i* = t) $\varepsilon_{xx}^{a}=0.025$, respectively. When the point defects concentration changes from 0.0 to 1.0 × 10^–7^, the Cr concentration increases from $c_{\mathrm{Cr}}^{c0}=0.77$ to $c_{\mathrm{Cr}}^{c1}=0.83$, and from $c_{\mathrm{Cr}}^{t0}=0.76$ to $c_{\mathrm{Cr}}^{t1}=0.84$ under the effects of compressive and tensile strains, as shown in Figure 5a,b. The concentration differences $\Delta c_{\mathrm{Cr}}^{i}=c_{\mathrm{Cr}}^{i1}-c_{\mathrm{Cr}}^{i0}$, of Cr in the α′ phase for compressive and tensile strains are $\Delta c_{\mathrm{Cr}}^{c}=0.06$ and $\Delta c_{\mathrm{Cr}}^{t}=0.08$, respectively.

With the increase in initial point defects concentration, the concentration difference of Cr in the α′ phase shows a very slight change. Therefore, point defects have little influence on the Cr concentration at low temperature (700 K), whether there is a normal strain or not. At 700 K, the effect of tensile strain on the concentration evolution is similar to that of compressive strain, while the compressive strain shows an obvious influence at 750 K (see Figure 4a,b). Under the effect of normal strain, the concentration change of vacancy is very small, as shown in Figure 5d. Similar to 750 K, the effect of the normal strain is very small for the concentration of point defects.

In order to further investigate the effect of point defects and strains at different temperatures, the equilibrium concentration of Cr, $c_{\mathrm{Cr}}^{\mathrm{eq}}$, in the α′ phase with temperature change is shown in Figure 6. It can be seen that the $c_{\mathrm{Cr}}^{\mathrm{eq}}$ decreases with the increased temperature with or without point defects. Under the influence of external strains, $c_{\mathrm{Cr}}^{\mathrm{eq}}$ is raised, as shown in Figure 6a,b. Zhu et al. studied the free energy change under the effects of external strain and found that the increase in strain makes the free energy decrease and leads to a higher Cr concentration $c_{\mathrm{Cr}}^{\mathrm{eq}}$ [20]. When the point defects concentration changes from 0.0 to 1.0 × 10^–7^, the $c_{\mathrm{Cr}}^{\mathrm{eq}}$ in the α′ phase shows an obvious decline under the effect of the compressive strain $\varepsilon_{xx}^{a}=-0.025$, as shown in Figure 6b. In addition, the Cr concentration differences among the three strain states become small. Therefore, the defects can reduce the difference of $c_{\mathrm{Cr}}^{\mathrm{eq}}$, especially under normal strain.

In the vacancy diffusion mechanism, the atom migrates via the position exchange with vacancy [46]. To study the evolution of vacancy and its influence on the phase separation under strain, the morphology evolution of the Cr-enriched α′ phase and concentration distributions of Cr and vacancy under $\varepsilon_{xx}^{a}=-0.025$ are shown in Figure 7. In the stage of nucleation and growth, as time changes from *t** = 0.001 to *t** = 2, the concentration of Cr increases from 0.35 to 0.44 with point defects, and to 0.41 without point defects, as shown in Figure 7d. At the same time, the Cr atoms cluster and form a small sized α′ phase, as shown in Figure 7a–d. The peak of vacancy concentration increases from 1.8 × 10^–7^ to 1.7 × 10^–4^ and moves from the initial boundary position to the center, as shown in Figure 7g–j. The rate of the concentration change of Cr in the α′ phase and vacancy is calculated by $\Delta c_{k}=\left(c_{k}^{t^{*}=2}-c_{k}^{t^{*}=0.001}\right)/c_{k}^{t^{*}=0.001}\;\left(k=\mathrm{Cr},\;V\right)$, $\Delta c_{\mathrm{Cr}}=0.257$ in the α′ phase, and $\Delta c_{V}=944.9$ for vacancy. The change rate of vacancy concentration is faster than that of the Cr-enriched α′ phase. The early stage evolution of vacancy provides more position to facilitate the mobility of Cr atoms, which makes the nucleation and growth of Cr-enriched nanoscale particles faster. With the increase in time, the vacancy is moving from the center to the interface of the α/α′ phases (see Figure 7k,l), and the exchange process of the vacancy and the Cr atoms promotes the growth of the Cr-enriched α′ phase [17]. In general, the cluster and mobility of point defects can induce a high diffusion path to promote the initial phase separation [47]. With the growth of the particle, the elongation of a Cr-enriched nanoscale particle becomes obvious under the compressive strain, and is weakened with the formation of the defects loop, as shown in Figure 7e,f. Similarly, the evolution of the Cr-enriched α′ phase under tensile strain is the same as that under compression strain.

### 3.3. Splitting of the α′ Phase with Defects Loop under Normal Strain

The splitting of nanoparticles under the influence of strain and temperature, and the relationship between the initial particle size and particle splitting are studied. The main factors of splitting are revealed by studying the elastic energy and interfacial energy. Figure 8 shows the concentration distribution in 2D and 3D for the Cr-enriched α′ phase at 700 K, 720 K and 750 K with $c_{V}^{i}=c_{I}^{i}=1\times 10^{-7}$ in an initial circle region under $\varepsilon_{xx}^{a}=-0.025$. At 700 K, the Cr concentration increases with time and then splits into two peaks, as shown in Figure 8a–d. It can be seen from the experiment that the particle splitting requires specific conditions [21], and that the phenomenon of particle splitting was also found during the growth and coarsening of γ′ precipitates in Ni-based superalloys [48,49], which is caused by the mismatch of the interface energy and the elastic energy. The particle splits into smaller ones to relax the elastic energy when its size reaches a critical value [48]. Therefore, the splitting is not a generic phenomenon. In the simulation of single particle splitting, the initial component of the particle is non-equilibrium, and with the growth of the particle, the particle splits [50]. When the concentration of the particle reaches a higher value or equilibrium, the particle itself does not continue to split, but the coarsening will continue [50]. Similarly, in our study, under a non-equilibrium initial concentration, the particle splits at low temperature and the splitting stops when the concentration reaches 0.54, as shown in Figure 8c. Therefore, the splitting is only a phenomenon under certain simulation conditions, and in our study it occurs at the initial phase separation.

At 720 K, the particle shows a concentration clustering and a tendency to split, as shown in Figure 8f,g, while the particle is merged again into one with an equilibrium concentration 0.862 in Figure 8h. Then the particle is elongated under the compressive strain. At 750 K, the concentration of Cr clusters and evolves into a particle with small size, then the particle grows up and elongates rapidly, as shown in Figure 8k–o. Because the atomic diffusion is faster at a high temperature, the lower interface energy and elastic energy in the center of particle will further inhibit the splitting of particle, as shown in Figure 8k,l, so that the Cr-enriched nanoparticle grows up and coarsens individually without splitting at 750 K.

In addition, when there is no point defect, the particle still splits at 700 K under a compressive strain $\varepsilon_{xx}^{a}=-0.025$, as shown in Figure 9a–e. When the strain decreases to $\varepsilon_{xx}^{a}=-0.019$, the split particles will be recombined, as shown in Figure 9h–j, which is similar to the state of 720 K (see Figure 8f–j) with a compressive strain $\varepsilon_{xx}^{a}=-0.025$. Therefore, the particles splitting can be eliminated by increasing the temperature and decreasing the strain. The results show that the point defects are not a determining factor for the splitting of particle, and that the particle splitting is affected by the interface energy and the strain energy [50]. When the ratio of the interfacial energy to the elastic energy is increased, the splitting becomes difficult [24]. Therefore, the relationship between the interfacial energy and the elastic energy needs to be discussed. With the increase in the applied strain, the α′ phase splits into two or more particles during its growth. Therefore, the particle’s splitting occurs under high strain.

To reveal the α′ phase splitting with a defects loop under normal strain, the dimensionless gradient energy density $E_{g}^{*}$, the dimensionless chemical free energy density $E_{c}^{*}$, the dimensionless elastic energy density $E_{e}^{*}$_,_ and the Cr concentration *c*_Cr_ distributions along the *y**-axis through the center of the α′ phase are shown in Figure 10. On the whole, the distribution of $E_{c}^{*}$ (see Figure 10b,f,j) is similar to that of *c*_Cr_ (see Figure 10a,e,i), while the distribution of $E_{e}^{*}$ inside of the α′ phase is contrary to that of *c*_Cr_.

At 700 K, the Cr concentration shows two peaks at *t** = 41, which means that the splitting of the α′ phase particle occurs. At the same time, the $E_{g}^{*}$ has a valley in the center position, as signed by the arrow in Figure 10d; while it is not matched with the increased $E_{e}^{*}$ in the center position, as shown by the arrows in Figure 10c, this energy that is different inside of the α′ phase causes the particle to split, as shown in Figure 8a–c. In the Ni-Al alloy with negative mismatch, the split of the γ′ precipitates results in the rapid increase in interface energy and a reduction in elastic strain energy, which is caused by the inverse precipitation of the γ phase in the center of the γ′ precipitates [48,50]. As the temperature increases, the change of $E_{g}^{*}$ and $E_{e}^{*}$ is gradually consistent in the center of particle at *t** = 41; as shown in Figure 10g,h,k,l, the particles coarsen independently without splitting.

In addition to the temperature, the initial particle size also affects the splitting of the particle in the single-particle system [50]. Figure 11 shows the concentration distribution in 2D and 3D for the Cr-enriched α′ phase at 700 K and 750 K in two-particle systems with $\varepsilon_{xx}^{a}=-0.025$ and $c_{V}^{i}=c_{I}^{i}=1\times 10^{-7}$, where the critical radius of the initial particle is *R=* 3.5 *l* at 700 K and 8 *l* at 750 K, respectively, as shown in Figure 11a,c. At 700 K, when the radius of the initial particle is less than *R* = 3.5 *l*, the particles coarsened independently without splitting under $\varepsilon_{xx}^{a}=-0.025$, as shown in Figure 11b. When the temperature increases to 750 K, the particles split at a larger particle radius of 8 *l*, as shown in Figure 11c,d. Therefore, the critical radius of particle splitting increases with the increase of the temperature. The split morphology of the two particles at 750 K is similar to that of single particles at 700 K in Figure 8e. Therefore, the particle size and their distance affect the splitting behavior in the multi-particle system [50]. Generally, when the size of the initial non-equilibrium particles is larger, the particles are easy to split at the early stage of phase separation under a higher normal strain (Appendix A).

## 4. Conclusions

The phase-field model coupled with defects and strain is developed to study the evolutions of point defects and α′ nanophase in the Fe-35at.% Cr alloys. The effects of the point defect and normal strain on the concentration evolution and the shape change of the nanoparticles are presented. It is shown that a coherent strain between the α/α′ phases is relaxed by forming a defects loop at the interface regions of the α/α′ phase, thus reducing the eigenstrain to weaken the deformation of the nanophase. The point defects enhance the effect of normal strain in accelerating the phase separation, which is obvious under the compressive strain at high temperature. Under the normal strain, the small initial radius of the Cr-enriched nanophase splits in the early-stage precipitation at 700 K, while the splitting disappears with the increased temperature and the decreased strains. In addition, the large initial size of the particles is more likely to split.

The equilibrium concentration of Cr in the α′ phase increases with the increased normal strain and the concentration of the point defects. However, under compression strain, the increased concentration of point defects can decrease the equilibrium concentration of Cr in the α′ phase. With the reconciliation of the point defects, the concentration differences become small for the strain-free, tensile, and compressive states. This study offers a unique perspective on the formation of the irradiation point defect loop in the phase separation alloy, and its influence on the evolution of nanoparticles under normal strain, and the relationship between point defects and normal strain are clarified.

## Figures and Tables

**Figure 1 nanomaterials-13-00456-f001:**
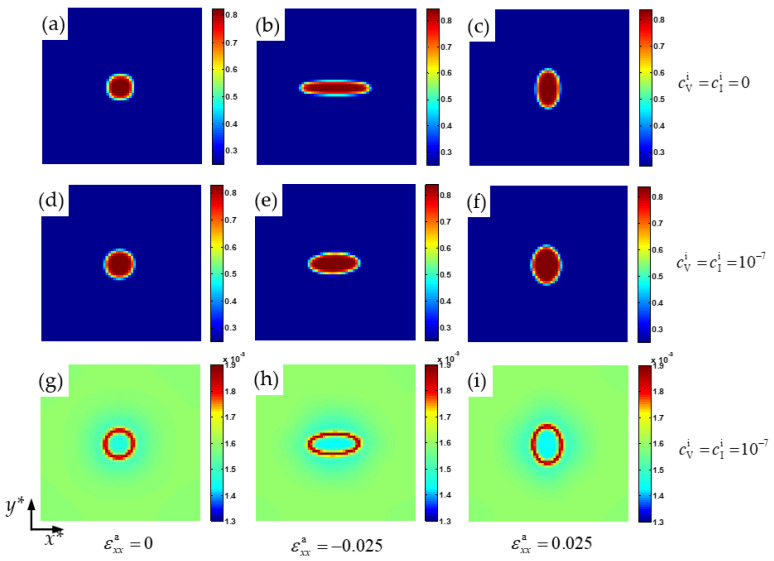
Morphology of the Cr-enriched α′ nanophase (**a**–**f**) and vacancy (**g**–**i**) at 750 K for *t** = 50. (**a**–**c**) $c_{V}^{i}=c_{I}^{i}=0$, (**d**–**i**) $c_{V}^{i}=c_{I}^{i}=1\times 10^{-7}$. (**a**,**d**,**g**) $\varepsilon_{xx}^{a}=0$, (**b**,**e**,**h**) $\varepsilon_{xx}^{a}=-0.025$, (**c**,**f**,**i**) $\varepsilon_{xx}^{a}=0.025$.

**Figure 2 nanomaterials-13-00456-f002:**
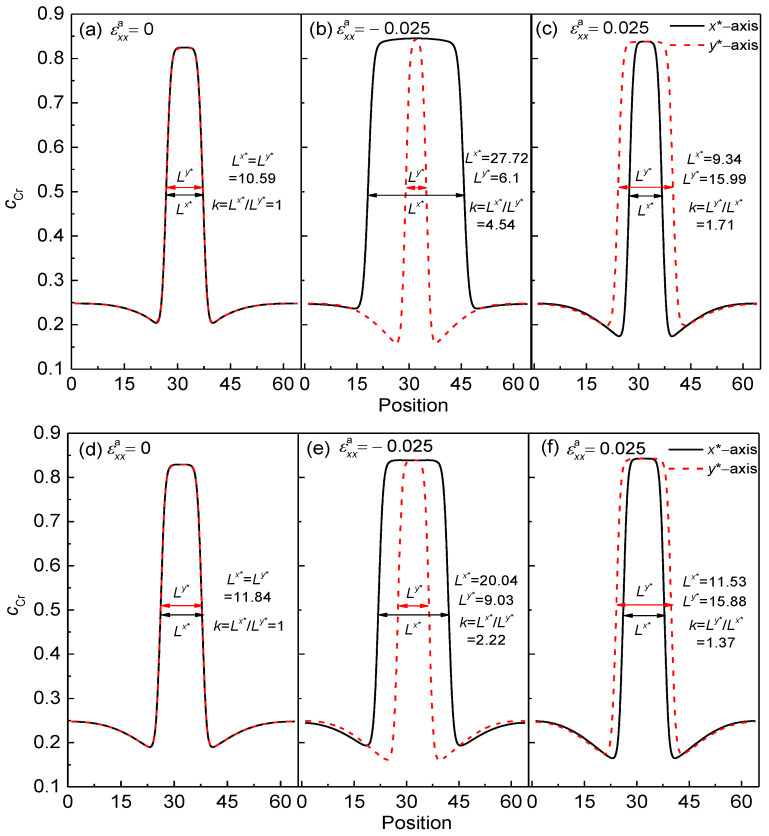
The concentration distributions of Cr in the α′ phase along *x** = 32 and *y** = 32 with and without normal strain at 750 K for *t** = 50. (**a**–**c**) $c_{V}^{i}=c_{I}^{i}=0$, and (**d**–**i**) $c_{V}^{i}=c_{I}^{i}=1\times 10^{-7}$.

**Figure 3 nanomaterials-13-00456-f003:**
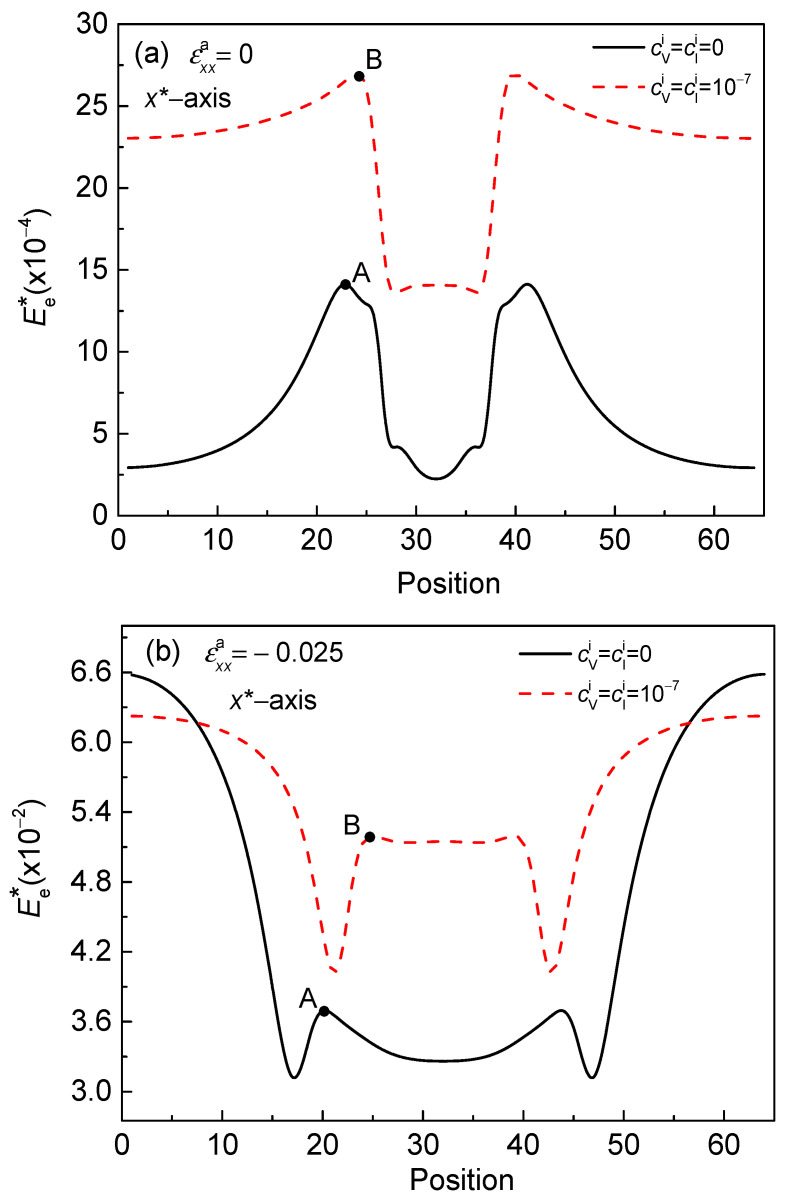
The dimensionless elastic energy density distribution through the center of α′ phase at *t** = 50 with and without point defects. (**a**) $\varepsilon_{xx}^{a}=0$, (**b**) $\varepsilon_{xx}^{a}=-0.025$, and (**c**) $\varepsilon_{xx}^{a}=0.025$.

**Figure 4 nanomaterials-13-00456-f004:**
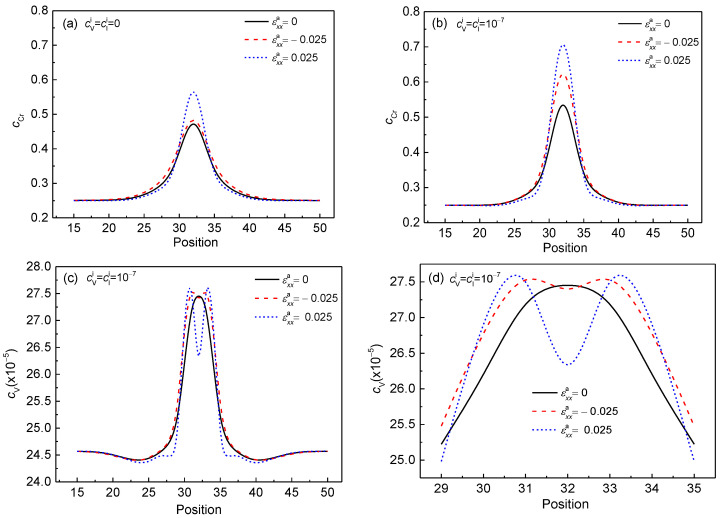
The concentration distributions of Cr and vacancy across the Cr-enriched α′ phase and vacancy at 750 K under different strains at *t** = 3. (**a**) $c_{V}^{i}=c_{I}^{i}=0$, (**b**–**d**) $c_{V}^{i}=c_{I}^{i}=1\times 10^{-7}$, and (**d**) local magnification of the concentration in Figure 4c.

**Figure 5 nanomaterials-13-00456-f005:**
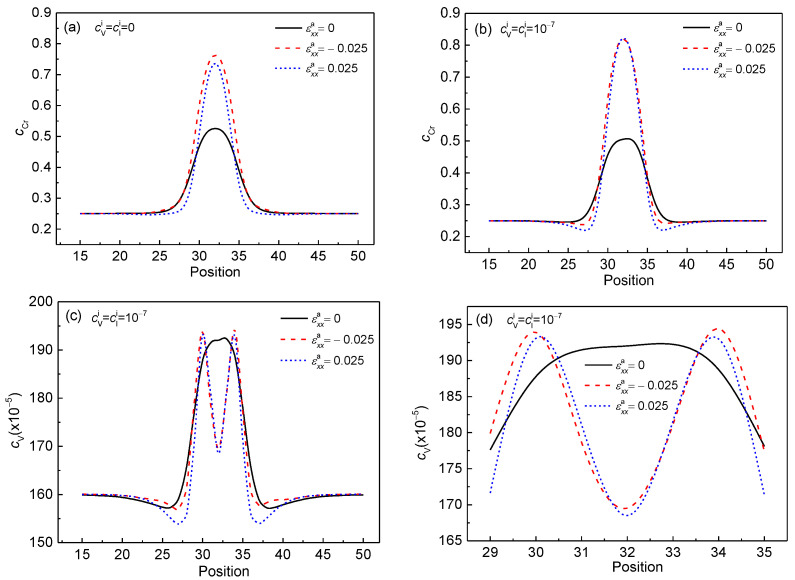
The concentration distributions of Cr and vacancy across the Cr-enriched α′ phase, and vacancy at 700 K under different strains at *t** = 71. (**a**) $c_{V}^{i}=c_{I}^{i}=0$, (**b**–**d**) $c_{V}^{i}=c_{I}^{i}=1\times 10^{-7}$, and (**d**) local magnification of the concentration in Figure 5c.

**Figure 6 nanomaterials-13-00456-f006:**
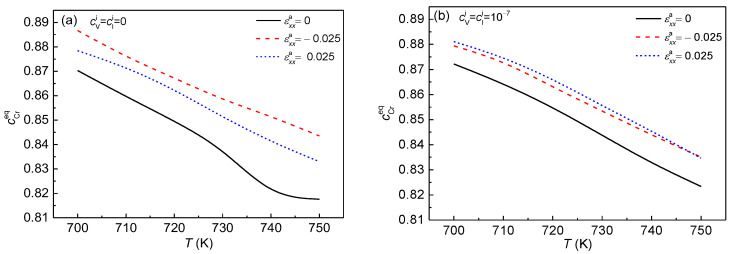
The equilibrium concentration of Cr in the α′ phase with temperature change at *t** = 791 under different strains. (**a**) $c_{V}^{i}=c_{I}^{i}=0$, (**b**) $c_{V}^{i}=c_{I}^{i}=1\times 10^{-7}$.

**Figure 7 nanomaterials-13-00456-f007:**
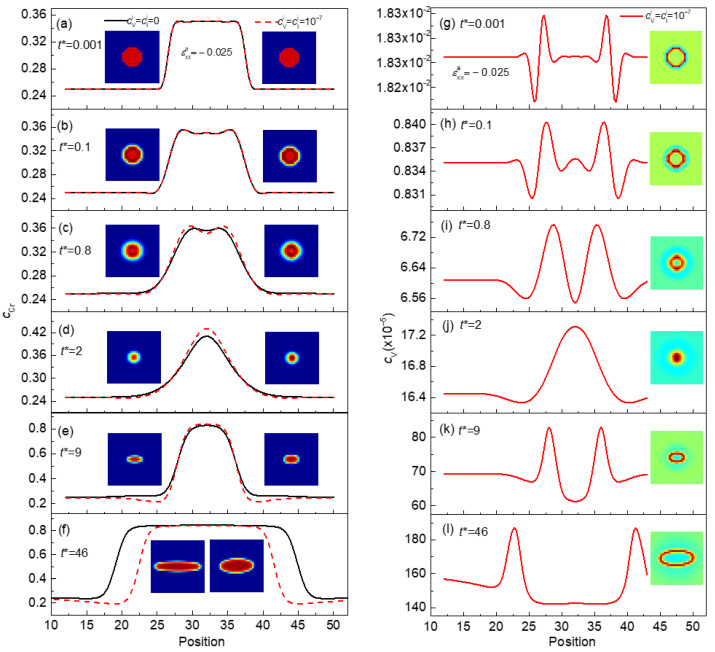
The morphology evolution of the Cr-enriched α′ phase and concentration distributions of Cr in the α′ phase with and without point defects (**a**–**f**), the morphology evolution and concentration distributions of vacancy (**g**–**l**) with $c_{V}^{i}=c_{I}^{i}=1\times 10^{-7}$, *T* = 750 K, $\varepsilon_{xx}^{a}=-0.025$.

**Figure 8 nanomaterials-13-00456-f008:**
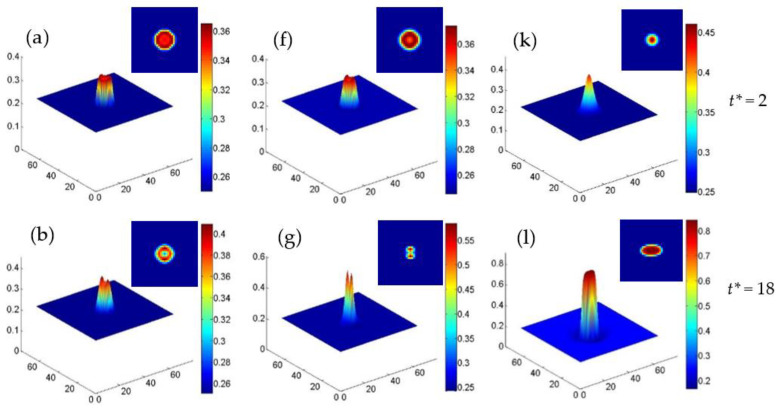
The concentration surface evolution of the Cr-enriched α′ phase from the initial circle regions with 35 at.% Cr and $c_{V}^{i}=c_{I}^{i}=1\times 10^{-7}$ under $\varepsilon_{xx}^{a}=-0.025$, (**a**–**e**) 700 K, (**f**–**j**) 720 K and (**k**–**o**) 750 K.

**Figure 9 nanomaterials-13-00456-f009:**
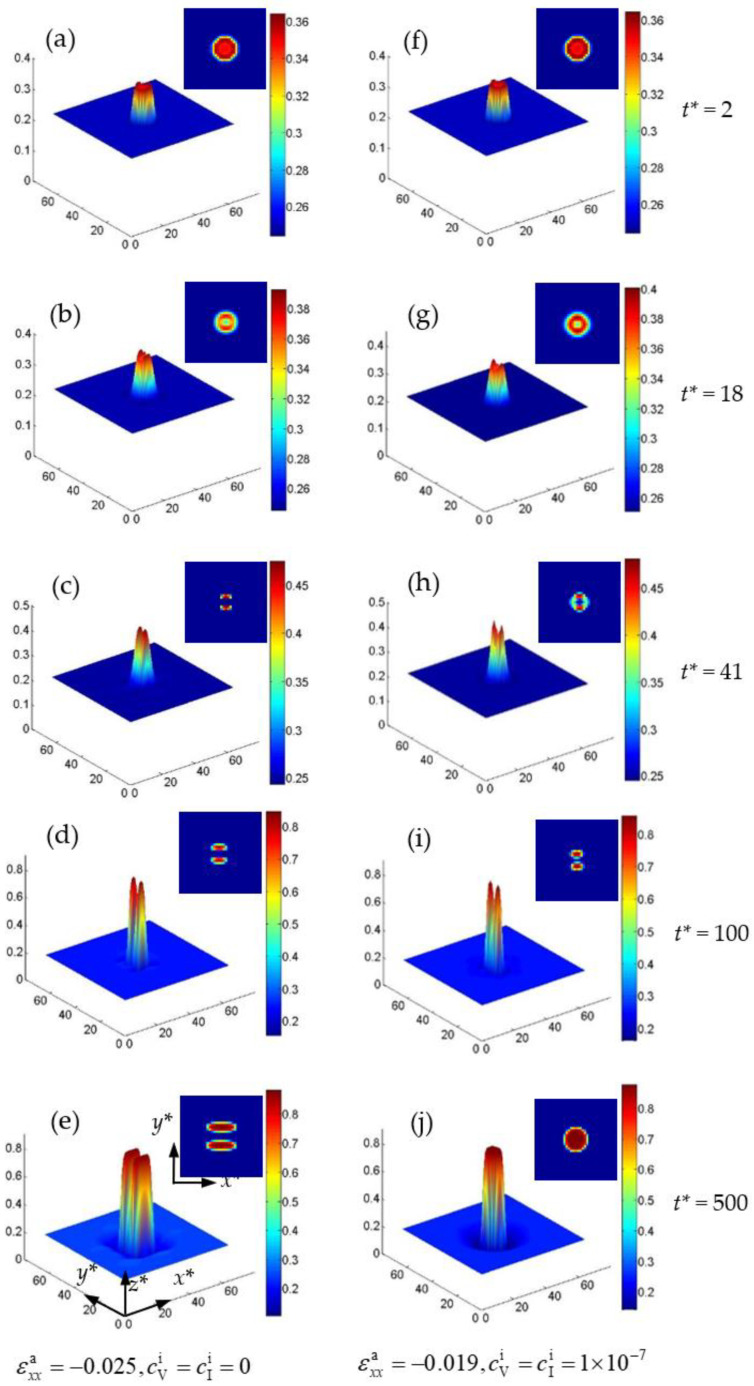
The concentration surface evolution of the Cr-enriched α′ phase from the initial circle regions with 35 at.% Cr at 700 K. (**a**–**e**) $\varepsilon_{xx}^{a}=-0.025,$ $c_{V}^{i}=c_{I}^{i}=0$, and (**f**–**j**) $\varepsilon_{xx}^{a}=-0.019,$ $c_{V}^{i}=c_{I}^{i}=1\times 10^{-7}$.

**Figure 10 nanomaterials-13-00456-f010:**
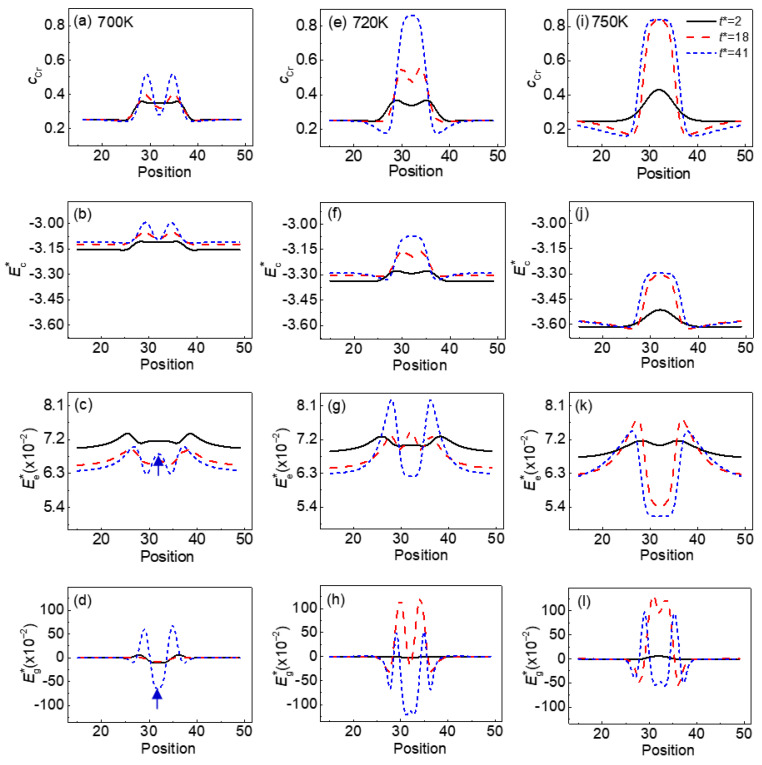
The dimensionless energy density and concentration distributions of Cr across of the Cr-enriched α′ phase with $c_{V}^{i}=c_{I}^{i}=1\times 10^{-7}$ at 700 K (**a**–**d**), 720 K (**e**–**h**) and 750 K (**i**–**l**) under $\varepsilon_{xx}^{a}=-0.025$. (**a**,**e**,**i**) concentration of Cr, (**b**,**f**,**j**) $E_{c}^{*}$, (**c**,**g**,**k**) $E_{e}^{*}$, (**d**,**h**,**l**) $E_{g}^{*}$.

**Figure 11 nanomaterials-13-00456-f011:**
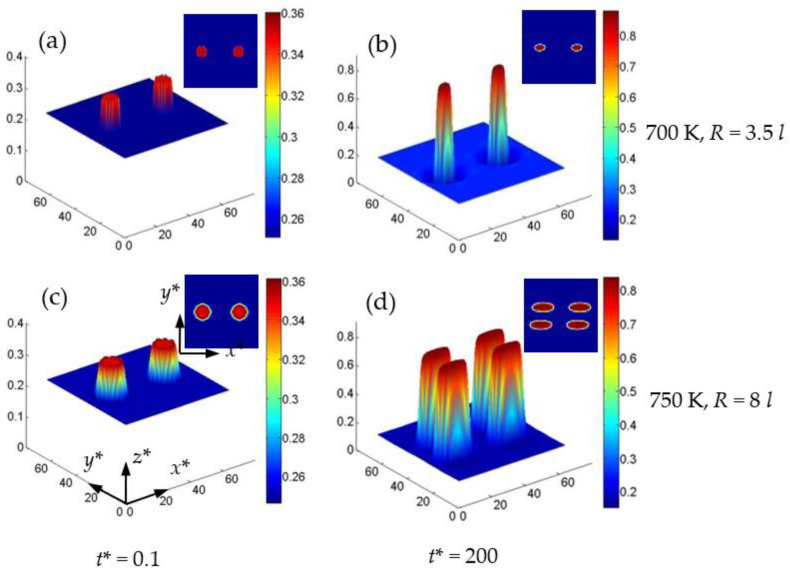
The concentration surface evolution of the Cr-enriched α′ phase from the initial circle regions with 35 at.% Cr under $\varepsilon_{xx}^{a}=-0.025$ and $c_{V}^{i}=c_{I}^{i}=1\times 10^{\times 7}$. (**a**,**b**) 700 K, *R* = 3.5 *l*, (**c**,**d**) 750 K, *R* = 8 *l*.

**Table 1 nanomaterials-13-00456-t001:** The concentration changes of Cr at 750 K under the effects of compressive and tensile strains.

	$c_{V}^{i}\;=\;c_{I}^{i}\;=\;0$			$c_{V}^{i}\;=\;c_{I}^{i}\;=\;1\times 10^{-7}$		
$\varepsilon_{xx}^{a}$	0	−0.025	0.025	0	−0.025	0.025
$c_{\mathrm{Cr}}$	$c_{\mathrm{Cr}}^{0}=0.48$	$c_{\mathrm{Cr}}^{c}=0.49$	$c_{\mathrm{Cr}}^{t}=0.58$	$c_{\mathrm{Cr}}^{0}=0.55$	$c_{\mathrm{Cr}}^{c}=0.64$	$c_{\mathrm{Cr}}^{t}=0.73$
$\Delta c_{\mathrm{Cr}}=c_{\mathrm{Cr}}^{i}-c_{\mathrm{Cr}}^{0}\;,(i=c,t)$	-	0.01	0.10	-	0.09	0.18
$\Delta c_{\mathrm{Cr}}/c_{\mathrm{Cr}}^{0}$	-	0.021	0.208	-	0.164	0.327

**Table 2 nanomaterials-13-00456-t002:** The concentration changes of Cr with and without defect at 700 K under the effects of compressive and tensile strains.

	$c_{V}^{i}\;=\;c_{I}^{i}\;=\;0$		$c_{V}^{i}\;=\;c_{I}^{i}\;=\;1\times 10^{-7}$	
$\varepsilon_{xx}^{a}$	−0.025	0.025	−0.025	0.025
$c_{\mathrm{Cr}}$	$c_{\mathrm{Cr}}^{c0}=0.77$	$c_{\mathrm{Cr}}^{t0}=0.76$	$c_{\mathrm{Cr}}^{c1}=0.83$	$c_{\mathrm{Cr}}^{t1}=0.84$
$\Delta c_{\mathrm{Cr}}^{i}=c_{\mathrm{Cr}}^{i1}-c_{\mathrm{Cr}}^{i0}\;,(i=c,t)$	$\Delta c_{\mathrm{Cr}}^{c}=0.06$		$\Delta c_{\mathrm{Cr}}^{t}=0.08$	

## Data Availability

The data presented in this study are available on request from the corresponding author.

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
