# Peer review of "Elastic Strain Relaxation of Phase Boundary of α′ Nanoscale Phase Mediated via the Point Defects Loop under Normal Strain"

_nanomaterials, 2023, doi:10.3390/nano13030456_

Round 1

Reviewer 1 Report

The paper deals with an interesting subject of irradiation induced point defects influence on precipitation kinetics in Fe-Cr-based steel. The presented simulation approach and the overall modelling strategy is well suited for such study and the results presented in the paper are well characterized and explained. Only one effect discussed in the paper is somewhat questionable. The particle splitting observed in the presented simulations seems to occur to the particle which are introduced at time t=0. Since these particles obviously have non-equilibrium shapes and composition the splitting may be a simulation transient rather then the generic effect. I would suggest authors to comment on this issue in the paper. In summary, I find the presented paper very interesting and recommend publishing it its current form.

Author Response

Ms. Ref. No.: nanomaterials-2079806

Title: Elastic strain relaxation of phase boundary of α′ nanoscale phase mediated via the point defects loop under normal strain

Dear Reviewer and Editor,
We are appreciating for your comments and suggestions on the manuscript nanomaterials-2079806 entitled with “Elastic strain relaxation of phase boundary of α′ nanoscale phase mediated via the point defects loop under normal strain”. We have fully considered the comments and revised the manuscript according to your comments. All the revisions are marked up using the “Track Changes” function in the resubmitted manuscript. All the responds are given one by one as following. The English of this article is edited in the MOPI.

Any questions about the revision please inform of us, thank you.

Yours sincerely,
Yongsheng Li

Reviewers' comments:

Reviewer 1:

The paper deals with an interesting subject of irradiation induced point defects influence on precipitation kinetics in Fe-Cr-based steel. The presented simulation approach and the overall modelling strategy is well suited for such study and the results presented in the paper are well characterized and explained. Only one effect discussed in the paper is somewhat questionable. The particle splitting observed in the presented simulations seems to occur to the particle which are introduced at time t=0. Since these particles obviously have non-equilibrium shapes and composition the splitting may be a simulation transient rather than the generic effect. I would suggest authors to comment on this issue in the paper. In summary, I find the presented paper very interesting and recommend publishing it its current form.

Response: It can be seen from the experiment that the particle splitting requires specific conditions [1], and that the phenomenon of splitting also found during the particle growing and coarsening in Ni-based superalloys [2,3], which is caused by the mismatch of the interface energy and the elastic energy. The particle splits into smaller ones to relax the elastic energy when its size reaches a critical value [2]. Therefore, the splitting is not a generic phenomenon. In the simulation of single particle splitting, the initial composition of the particle is non-equilibrium, and with the growth of the particle, the particle splits [4]. When the concentration of particle reaches a higher value or equilibrium, the particle itself does not continue to split, but the coarsening will continue [4]. Similarly, in our study, under a non-equilibrium initial concentration, the particle splits at low temperature and the splitting stops when the concentration reaches 0.54, as shown in Figs 8c. Therefore, the splitting is only a phenomenon under certain simulation conditions, and in our study it occurs at the initial phase separation.

Reference:

  1. Doi, M.; Miyazaki, T.; Wakatsuki, T. The effect of elastic interaction energy on the morphology of γ′ precipitates in nickel-based alloys. Mater Sci Eng. 1984, 67, 247–253. https://doi.org/10.1016/0025-5416(84)90056-9.
  2. Banerjee, D.; Banerjee, R.; Wang, Y. Formation of split patterns of γ′ precipitates in Ni-Al via particle aggregation. Scr Mater. 1999, 41, 1023–1030.

https://doi.org/10.1016/S1359-6462(99)00223-7.

  1. Yoo, Y.S.; Yoon, D.Y. Henry, M.F. The effect of elastic misfit strain on the morphological evolution of γ′-precipitates in a model Ni-base superalloy. Met Mater. 1995, 1, 47–61. https://doi.org/10.1007/BF03055324.
  2. Liu, L.; Chen, Z.; Wang, Y.X. Elastic strain energy induced split during precipitation in alloys. J Alloy Compd. 2016, 661, 349–356. https://doi.org/10.1016/j.jallcom.2015.11.201.

Reviewer 2 Report

The present work is under way to become an interesting contribution to the field, but it is not acceptable in its present state. It is very difficult to follow and is in need of an extensive overhaul before publication.

The temporal evolution of the system is unclear. Just snapshots from different dimensionless times are presented.

The matrix and precipitate compositions are unclear. On line 201 at-% Cr in the precipitate is stated to be 35, which doesn’t match Figs. 1-2. Then again in section 3.2 (Figs. 4-5) the Cr mole fraction profiles differ markedly from the earlier results.

Lots and lots of numerical results are given in the text, rather than being presented in tables and/or figures.

The mesh seems a bit coarse.

It seems that mole fraction is used throughout, but the words “composition” or “compositions” are used, e.g. “…the compositions of solution atoms and point defects..” (lines 106-107).

The units of E_V^f and E_I^f in Eq. (11) are wrong.

No phase-field variables are used, so this is not phase-field modelling.

Author Response

Ms. Ref. No.: nanomaterials-2079806

Title: Elastic strain relaxation of phase boundary of α′ nanoscale phase mediated via the point defects loop under normal strain

Dear Reviewer and Editor,
We are appreciating for your comments and suggestions on the manuscript nanomaterials-2079806 entitled with “Elastic strain relaxation of phase boundary of α′ nanoscale phase mediated via the point defects loop under normal strain”. We have fully considered the comments and revised the manuscript according to your comments. All the revisions are marked up using the “Track Changes” function in the resubmitted manuscript. All the responds are given one by one as following. The English of this article is edited in the MOPI.

Any questions about the revision please inform of us, thank you.

Yours sincerely,
Yongsheng Li

Reviewers' comments:

Reviewer 2:

The present work is under way to become an interesting contribution to the field, but it is not acceptable in its present state. It is very difficult to follow and is in need of an extensive overhaul before publication.

The temporal evolution of the system is unclear. Just snapshots from different dimensionless times are presented.

Response: Fig. 7 shows the evolution of the morphology with time under compressive strain, and a gif of the evolution of the Cr-enriched α′ phase and point defect under compressive strain is added in the attachment. The snapshots from different dimensionless times are presented to compare the evolution characteristics and laws of α′ phase with point defect under different stress states.

The matrix and precipitate compositions are unclear. On line 201 at-% Cr in the precipitate is stated to be 35, which doesn’t match Figs. 1-2. Then again in section 3.2 (Figs. 4-5) the Cr mole fraction profiles differ markedly from the earlier results.

Response: The ferrite phase is separated into Cr-enriched α′ precipitated phase and Fe-enriched α matrix phase, which have the same structure. The 35at.% Cr is the nominal composition of the alloy, and the initial state of simulation is 35at.% Cr in a circle region with a radius of 5 l and 25 at.% Cr in the other regions of the simulation cell. Over time, the phase separation will happen in Fe-Cr alloy which will eventually bring the concentration into equilibrium under the effects of temperature and stress. When t* = 50, the concentration of Cr reached the equilibrium value 0.84 and 0.83 with and without strain, as shown in Figs. 1-2. Similarly, in Figs. 4 and 5, the particles are in the stage of growing and coarsening.

Lots and lots of numerical results are given in the text, rather than being presented in tables and/or figures.

Response: We have compiled the data analyzed in Figs. 4 and 5 into tables 1 and 2, which are as follows:

Table 1 The concentration changes of Cr at 750 K under the effects of compressive and tensile strains.

0

-0.025

0.025

0

-0.025

0.025

-

0.01

0.10

-

0.09

0.18

-

0.021

0.208

-

0.164

0.327

Table 2 The concentration changes of Cr with and without defect at 700 K under the effects of compressive and tensile strains.

-0.025

0.025

-0.025

0.025

The mesh seems a bit coarse.

Response: In the multi-particle system, the fine evolution characteristics of the particles themselves are not easy to capture due to the connection coarsening during the evolution of the particles. Therefore, in order to capture the evolution characteristics of nanoparticles more clearly, the single-particle system is used. For a single-particle system, a  system is used to save time. In addition, the colormap is used to show more clearly the transition of interface among α phase, α′ phase and point defects loop.

It seems that mole fraction is used throughout, but the words “composition” or “compositions” are used, e.g. “…the compositions of solution atoms and point defects.” (lines 106-107).

Response: The concentration of Cr and vacancy is expressed by the atomic percentage at.%. The compositions have been replaced by concentration in this article.

The units of E_V^f and E_I^f in Eq. (11) are wrong.

Response: In this article, in order to unify the units on both sides of the Eq. (11), the units of and  in Eq. (11) are J/mol, it can be transformed with eV by the following formula: 1eV=1×1.6×10-19NA J/mol.

No phase-field variables are used, so this is not phase-field modelling.

Response: Since α′ phase and α phase have the same structure, the difference in structure between the two phases is ignored, and only the difference in concentration is considered. Therefore, the order parametric equation in the phase field is not considered. The field variations in the phase-field model are described by the composition field of Cr , vacancy , and interstitial atoms through the Cahn–Hilliard diffusion equation which be used to discuss the evolution of Cr-enriched α′ and point defect in this article.
